# The entorhinal-DG/CA3 pathway in the medial temporal lobe retains visual working memory of a simple surface feature

Weizhen Xie[1,2,3]\*, Marcus Cappiello[2], Michael A Yassa[4], Edward Ester[5], Kareem A Zaghloul[1], Weiwei Zhang[2]\*

[1]Surgical Neurology Branch, National Institute of Neurological Disorders and Stroke, Bethesda, United States; [2]Department of Psychology, University of California, Riverside, Riverside, United States; [3]Department of Psychology, University of Maryland, College Park, United States; [4]Center for the Neurobiology of Learning and Memory, School of Biological Sciences, University of California, Irvine, Irvine, United States; [5]Department of Psychology, University of Nevada, Reno, United States

\*For correspondence:
weizhen.xie@nih.gov (WX);
weiwei.zhang@ucr.edu (WZ)

**Competing interest:** The authors declare that no competing interests exist.

**Abstract** Classic models consider working memory (WM) and long-term memory as distinct mental faculties that are supported by different neural mechanisms. Yet, there are significant parallels in the computation that both types of memory require. For instance, the representation of precise item-specific memory requires the separation of overlapping neural representations of similar information. This computation has been referred to as pattern separation, which can be mediated by the entorhinal-DG/CA3 pathway of the medial temporal lobe (MTL) in service of long-term episodic memory. However, although recent evidence has suggested that the MTL is involved in WM, the extent to which the entorhinal-DG/CA3 pathway supports precise item-specific WM has remained elusive. Here, we combine an established orientation WM task with high-resolution fMRI to test the hypothesis that the entorhinal-DG/CA3 pathway retains visual WM of a simple surface feature. Participants were retrospectively cued to retain one of the two studied orientation gratings during a brief delay period and then tried to reproduce the cued orientation as precisely as possible. By modeling the delay-period activity to reconstruct the retained WM content, we found that the anterior-lateral entorhinal cortex (aLEC) and the hippocampal DG/CA3 subfield both contain item-specific WM information that is associated with subsequent recall fidelity. Together, these results highlight the contribution of MTL circuitry to item-specific WM representation.

## Editor's evaluation

This useful study highlights the contribution of the medial temporal lobe (MTL), and the DG/CA3 hippocampal pathway in particular, to neural activity during the working memory delay period. The evidence supporting this is compelling, using diverse state-of-the-art approaches to neural data analysis and relating it to behavioural data. The work will be of significant interest to neuroscientists specialising in the research area of human working memory.

## Introduction

Working memory (WM) or short-term memory actively retains a small amount of information to support ongoing mental processes (*Baddeley, 2012*). This core mental faculty relies upon distributed

brain regions (*Christophel et al., 2017*; *Eriksson et al., 2015*), ranging from lower-level sensory areas (*Harrison and Tong, 2009*; but see *Bettencourt and Xu, 2016*) to higher-level frontoparietal networks (*Bettencourt and Xu, 2016*; *Ester et al., 2015*; *Todd and Marois, 2004*; *Xu and Chun, 2006*). This distributed neocortical network, however, often does not involve the medial temporal lobe (MTL), which is traditionally attributed to long-term episodic memory (*Eichenbaum et al., 2007*; *Squire and Zola-Morgan, 1991*). This distinction is grounded in the separation between WM and long-term memory in classic models (*Atkinson and Shiffrin, 1968*; *Norris, 2017*) and in early MTL lesion case studies (*Milner et al., 1968*; *Scoville and Milner, 1957*). Yet, this classic view is not free of controversy. A growing body of research has suggested that the MTL is involved in tasks that rely on information maintained in WM (*Boran et al., 2022*; *Boran et al., 2019*; *Hannula and Ranganath, 2008*; *Johnson et al., 2018*; *Kamiński et al., 2017*; *Kornblith et al., 2017*; *Libby et al., 2014*; *Liu et al., 2020*; *Rissman et al., 2008*; *Xie et al., 2023a*; *Xie and Zaghloul, 2021*). Furthermore, MTL lesions can disrupt WM task performance (*Goodrich et al., 2019*; *Koen et al., 2017*; *Olson et al., 2006*; *Warren et al., 2014*; *Xie et al., 2023a*). Despite these recent findings, however, major theories have not considered the MTL as a mechanism underlying WM (*Jeneson and Squire, 2012*; *Sreenivasan and D'Esposito, 2019*). First, it is unclear what computational process of the MTL is involved in WM (*Sreenivasan and D'Esposito, 2019*). Furthermore, the MTL tends to engage more in a WM task when long-term memory becomes relevant, for example when task loads are higher (*Boran et al., 2022*; *Boran et al., 2019*; *Rissman et al., 2008*) or when task stimuli are complex (*Barense et al., 2007*; *Borders et al., 2022*; *Kamiński et al., 2017*; *Kornblith et al., 2017*; *Libby et al., 2014*; *Liu et al., 2020*). As a result, contributions of the MTL to WM are often deemed secondary (*Jeneson and Squire, 2012*; *Sreenivasan and D'Esposito, 2019*).

Clarifying this issue requires specifying how the MTL contributes to WM representation and the extent to which this contribution holds even when WM task demand is minimized. Although WM and long-term memory are traditionally considered separate mental faculties, the functional parallels in both types of memory suggest potential shared neural mechanisms (*Beukers et al., 2021*; *Cowan, 2001*; *Nee and Jonides, 2008*; *Ruchkin et al., 2003*). For example, the ability to retain precise item-specific memory would require the computation to distinguish neural representations of similar information – a process known as pattern separation (*Marr, 1971*). This aspect of long-term memory is widely thought to emerge from various properties of the MTL's entorhinal-DG/CA3 pathway (*Aimone et al., 2011*; *Bakker et al., 2008*; *Cappiello et al., 2016*; *Ekstrom and Yonelinas, 2020*; *Korkki et al., 2021*; *Leal and Yassa, 2018*; *Marr, 1971*; *Reagh and Yassa, 2014*; *Yassa and Stark, 2011*), such as abundant granule cells and strong inhibitory interneurons in the hippocampal DG, as well as powerful mossy fiber synapses between the DG and CA3 subfields (*Aimone et al., 2011*; *Sahay et al., 2011*). These properties make it possible to enable sparse coding to ensure a sufficient representational distance among similar information (*Rolls, 2016*; *Rolls, 2013*). As these hippocampal substructures communicate with other neocortical areas via the entorhinal cortex (*Aimone et al., 2011*; *Leal and Yassa, 2018*), there is a proposed gradian of pattern separation along the entorhinal-DG/CA3 pathway to support item-specific long-term episodic memory (*Reagh and Yassa, 2014*). These ideas are supported by evidence based on animal and human behaviors (*Burke et al., 2011*; *Hunsaker et al., 2008*; *Ryan et al., 2012*), electrophysiological recordings (*Leutgeb et al., 2007*; *Lohnas et al., 2018*; *Sakon and Suzuki, 2019*), and human fMRI (*Bakker et al., 2008*; *Leal and Yassa, 2018*; *Montchal et al., 2019*; *Reagh and Yassa, 2014*). However, the extent to which the entorhinal-DG/CA3 pathway is involved in WM, especially in humans other than animal models (*Gilbert and Kesner, 2006*), has remained unknown.

Several challenges faced in past research may add to this uncertainty. For example, it is difficult to infer signals from MTL substructures, especially those within the hippocampus, based on human fMRI using a standard spatial resolution (*Bettencourt and Xu, 2016*; *Ester et al., 2015*) or intracranial direct recording with limited electrode coverage (*Boran et al., 2019*; *Johnson et al., 2018*; *Kamiński et al., 2017*; *Kornblith et al., 2017*). Furthermore, the use of complex task designs with multiple memory items (*Borders et al., 2022*) might also be suboptimal to reveal item-specific WM information in MTL subregions without being too taxing on the WM storage limit. To investigate these issues, here, we leverage an established retro-cue orientation WM task (*Bettencourt and Xu, 2016*; *Ester et al., 2015*; *Harrison and Tong, 2009*) and a high-resolution fMRI protocol to test the key prediction that the MTL's entorhinal-DG/CA3 pathway retains item-specific WM information of a simple

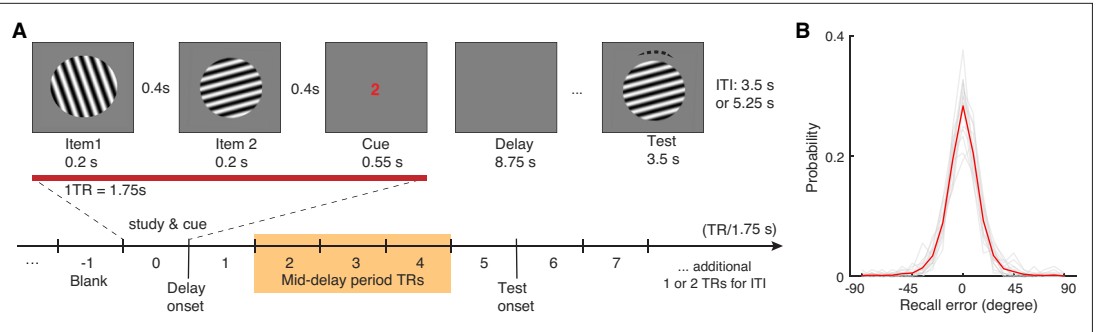

**Figure 1.** Visual WM task and participants' task performance. (**A**) During fMRI scanning, participants were directed to retain the orientation of a cued grating stimulus from two sequentially presented grating stimuli (item 1 vs 2). After a short retention interval, they tried to reproduce the cued orientation grating as precisely as possible. (**B**) Participants' task performance was high and mostly driven by the fidelity of the retained visual WM content. Each gray trace represents a participant's recall probability in the feature space (−90 to 90 degrees). The red trace represents across-subject average. TR = MR repetition time; ITI = inter-trial interval. The shaded area in (**A**) highlights the middle 3 TRs of the delay period. See *Figure 1—figure supplement 1* for additional details.

The online version of this article includes the following figure supplement(s) for figure 1:

**Figure supplement 1.** Behavioral task design and data in the current study across participants.

surface feature. In this task, participants are directed to retain the orientation information of a cued stimulus from two sequentially presented orientation gratings (separated by >20°; *Figure 1A*). After a short delay (5 TRs; 1TR = 1.75 s), they try to reproduce the cued orientation grating as precisely as possible using the method of adjustment. As participants are retrospectively cued to retain only one item during the delay, they are expected to encode both items but then only keep one in mind during the delay period. This design imposes a task demand on the observer to correctly remember the cued orientation while resisting the interference from the internal representations of other similar orientation gratings. The retention of information selected after encoding over a short delay has been considered a hallmark of WM (*Lorenc et al., 2021*; *Panichello and Buschman, 2021*), regardless of the presence or absence of sustained neural activation (*Lundqvist et al., 2018*; *Rose et al., 2016*). If the MTL's entorhinal-DG/CA3 pathway indeed supports this function, it is expected that the recorded delay-period activity should contain more information about the cued item, as compared with the uncued item, even though both items are initially remembered with an equal likelihood (*Bettencourt and Xu, 2016*; *Ester et al., 2015*; *Harrison and Tong, 2009*). If, however, information about the cued and uncued items is equally present during the delay period, the MTL may play a limited role in the representation of task-relevant information in WM but more during the initial encoding.

## Results

Participants' memory performance is quantified as recall error – the angular difference between the reported and the actual orientations of the cued item (*Zhang and Luck, 2008*). As the effective memory set size is low at one memory item, participants' performance is high with an average absolute recall error of 12.01°±0.61° (mean ± s.e.m.). Furthermore, the recall error distribution is centered around 0° with most absolute recall errors smaller than 45° (~97% trials; *Figure 1B*). These behavioral data suggest that participants in general have remembered high-fidelity orientation information of the cued item during the delay period.

### Fine discrimination of remembered WM content in the MTL

Of primary interest, we examined whether precise orientation information of the cued item is retained during WM retention in anatomically defined MTL regions of interest (ROIs; *Figure 2A*), including the entorhinal cortex (anterior-lateral, aLEC and posterior-medial, pMEC), the perirhinal cortex, para-hippocampus, and hippocampal DG/CA3, CA1, subiculum, as defined in the previous studies (*Montchal et al., 2019*; *Reagh et al., 2017*). Additionally, we chose the amygdala as a theoretically irrelevant but adjacent control region, because the involvement of the amygdala for emotionally neutral orientation information is expected to be minimal (*Iwai et al., 1990*). This allows us to gauge

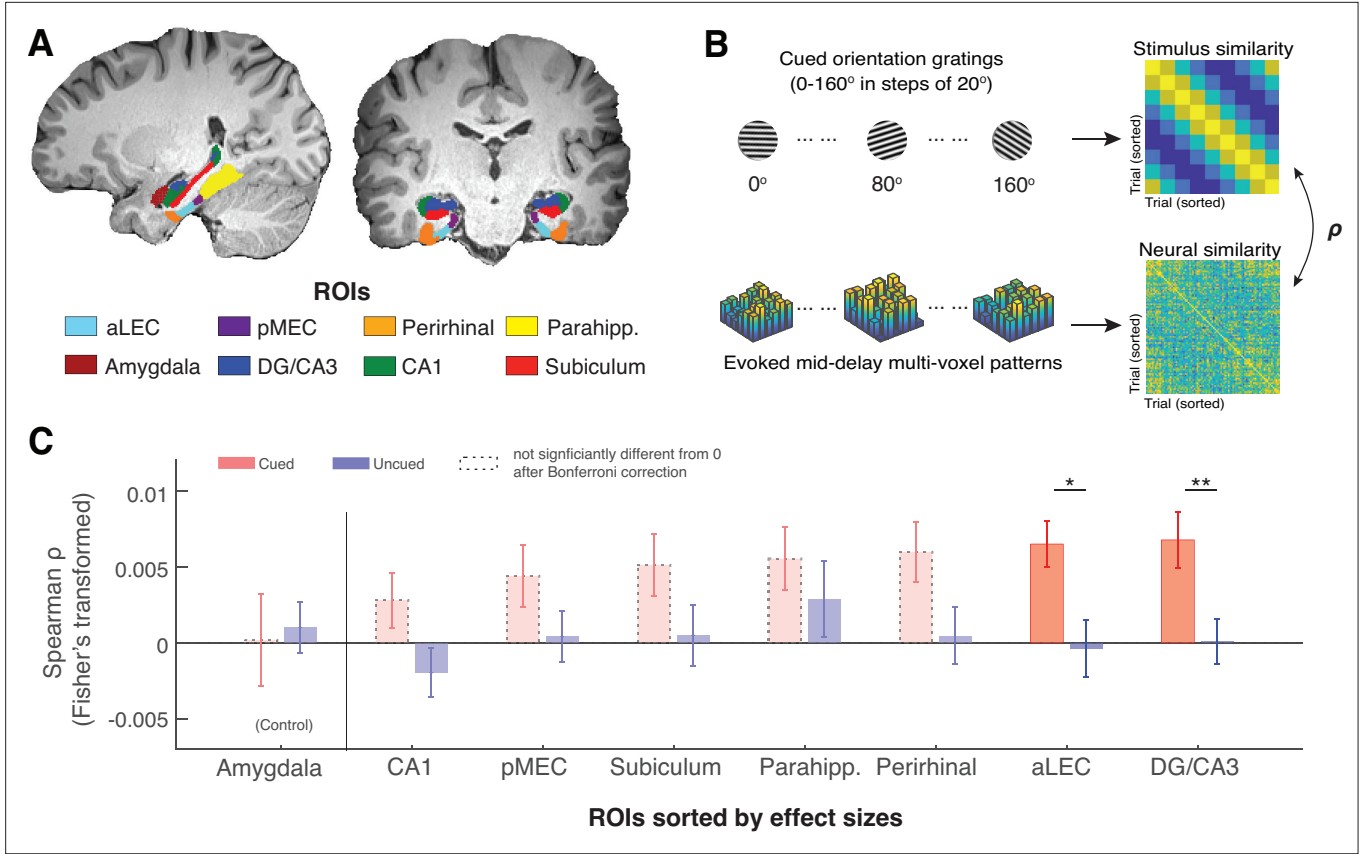

**Figure 2.** The MTL retains item-specific WM information revealed by stimulus-based representational similarity analysis. (**A**) MTL ROIs are parcellated based on previous research (*Montchal et al., 2019*; *Reagh et al., 2017*). The amygdala is chosen as an adjacent control region. (**B**) For each ROI, we examined the extent to which the evoked multi-voxel pattern during the mid-delay period could keep track of the feature values among different WM items. Specifically, we correlated the similarity in evoked neural patterns during the WM delay period separately with the feature similarity of every two cued items and with that of every two uncued items. The rationale is that if a brain region contains item-specific information to allow fine discrimination of different items, the evoked neural patterns should keep track of the feature similarity of these items (*Kriegeskorte and Wei, 2021*). (**C**). Across ROIs, we find that this prediction is supported by data from the aLEC and DG/CA3, which show a larger effect size in the association between neural and stimulus similarity patterns based on the cued item as compared with the uncued item. Error bars represent the standard error of the mean (s.e.m.) across participants. *p<0.05 and **p<0.01 for the comparison of the results based on cued versus uncued items; aLEC = anterior-lateral entorhinal cortex; pMEC = posterior-medial entorhinal cortex; parahipp. = parahippocampus. Results from detailed statistical tests are summarized in *Supplementary file 1a*.

The online version of this article includes the following figure supplement(s) for figure 2:

**Figure supplement 1.** Voxel responses in an example ROI (aLEC) for different remembered stimuli from one example subject.

**Figure supplement 2.** Across-region neural similarity analysis using the combined aLEC-DG/CA3 as an MTL seed region, the superior parietal lobule (SPL) ROI as a benchmark region, and the amygdala as a control region.

the observations in MTL ROIs while controlling for the signal-to-noise ratio in fMRI blood-oxygenation-level-dependent (BOLD) signals in deep brain structures.

As recent neural theories of WM have proposed that information retained in WM may not rely on sustained neural activation (*Ester et al., 2015*; *Kamiński and Rutishauser, 2020*; *Rose et al., 2016*), we inspected how the multivoxel activity pattern in each subject-specific ROI is correlated with the retained WM content predicted by the cued orientation gating (*Figure 2B*). We found that certain voxels in an ROI could respond more strongly to a particular cued orientation, even when the average BOLD activity across voxels does not show preferred coding for a certain orientation (see an example in *Figure 2—figure supplement 1*). We then assessed the consistency of these stimulus-related multi-voxel activity patterns in the MTL and the amygdala control region based on stimulus-based representational similarity analysis. In this analysis, we correlated the angular similarity of every pair of cued orientation gratings with the similarity of the evoked BOLD patterns in these trials. The rationale

is that if orientation information is retained within an ROI, the recorded neural data should track the relative angular distance between any two cued orientation gratings (hence fine discrimination *Kriegeskorte and Wei, 2021*). Informed by the previous research (*Ester et al., 2015*; *Harrison and Tong, 2009*), we performed this analysis using the raw fMRI BOLD signals from the middle 3TRs out of the 5-TR retention interval to minimize the contribution of sensory process or anticipated retrieval, hence maximizing the inclusion of neural correlates of WM retention (*Postle et al., 2000*).

In line with our prediction, we found that stimulus similarity for the cued item was significantly correlated with neural similarity across trials as compared with the null in both the aLEC (t(15) = 4.29, p=6.48e-04, $p_{Bonferroni}$ = 0.0052, Cohen's d=1.11, $p_{boostrap}$ <0.001) and the hippocampal DG/CA3 (t(15) = 3.64, p=0.0024, $p_{Bonferroni}$ = 0.019, Cohen's d=0.94, $p_{boostrap}$ <0.001; *Figure 2C*). In contrast, stimulus similarity for the uncued item across trials could not predict these neural similarity patterns in these regions as compared with the null (aLEC: t(15) = –0.20, p=0.85, Cohen's d=–0.05; DG/CA3: t(15) = 0.06, p=0.95, Cohen's d=0.02; $p_{boostrap}$'s>0.50). Furthermore, the evoked neural similarity patterns in these regions were significantly more correlated with the cued item as compared with the uncued item (aLEC: t(15) = 2.66, p=0.018, Cohen's d=0.69, $p_{boostrap}$ = 0.015; DG/CA3: t(15) = 3.64, p=0.0024, Cohen's d=0.94, $p_{boostrap}$ = 0.0016). While the rest of the MTL showed similar patterns, we did not obtain significant evidence in other MTL ROIs following the correction of multiple comparisons (see *Supplementary file 1a* for full statistics). Furthermore, neural evidence related to the cued item in the aLEC and DG/CA3 was significantly stronger than that in the amygdala control ROI. This was supported by a significant cue (cued vs. uncued) by region (combined aLEC-DG/CA3 vs. amygdala) interaction effect on the correlation between stimulus and neural similarity patterns (F(1, 15)=4.97, p=0.042, $p_{boostrap}$ = 0.036). Together, these results suggest that delay-period activity patterns in the entorhinal-DG/CA3 pathway are associated with retrospectively selected task-relevant information, implying the presence of item-specific WM representation in these subregions.

## Reconstruction of item-specific WM information based on inverted encoding modeling

To directly reveal the item-specific WM content, we next modeled the multivoxel patterns in subject-specific ROIs using an established inverted encoding modeling (IEM) method (*Ester et al., 2015*). This method assumes that the multivoxel pattern in each ROI can be considered as a weighted summation of a set of orientation information channels (*Figure 3A*). By using partial data to train the weights of the orientation information channels and applying these weights to an independent hold-out test set, one can reconstruct the assumed orientation information channels to infer item-specific information for the remembered item – operationalized as the resultant vector length of the reconstructed orientation information channel normalized at 0° reconstruction error (*Figure 3—figure supplement 1*). As this approach verifies the assumed information content based on observed neural data, its results can be efficiently computed and interpreted within the assumed model even when the underlying neuronal tuning properties are unknown (*Ester et al., 2015*; *Sprague et al., 2018*). This approach, therefore, complements the model-free similarity analysis by linking representational geometry embedded in the neural data with item-specific information under a model-based framework (*Kriegeskorte and Wei, 2021*; *Xie et al., 2023b*). On the basis of this method, previous research has revealed item-specific WM information in distributed neocortical areas, including the parietal, frontal, and occipital-temporal areas (*Bettencourt and Xu, 2016*; *Ester et al., 2015*; *Rademaker et al., 2019*; *Sprague et al., 2016*), which are similar to those revealed by other multivariate classification methods (e.g. support vector machine, SVM, *Ester et al., 2015*). We have also replicated these IEM effects in the current dataset (*Figure 3—figure supplement 2*).

Moving beyond these well-established observations in distributed neocortical structures, we found that the amount of reconstructed item-specific information for the cued item during WM retention was also significantly greater than chance level in two anatomically defined MTL subregions, aLEC (t(15) = 4.41, p=5.07e-04, $p_{bonferroni}$ = 0.0041, Cohen's d=1.14, $p_{boostrap}$ <0.001) and the hippocampal DG/CA3 (t(15) = 4.73, p=2.68e-04, $p_{bonferroni}$ = 0.0021, Cohen's d=1.22, $p_{boostrap}$ <0.001; *Figure 3B*). These effects were specific to the maintenance of the cued item, as information related to the uncued item was not statistically different from chance (aLEC: t(15) = –0.35, p=0.74, Cohen's d=–0.09; DG/CA3: t(15) = 0.66, p=0.52, Cohen's d=0.17; $p_{boostrap}$'s>0.50) and was significantly less than that for the cued item (aLEC: t(15) = 2.75, p=0.015, Cohen's d=0.71, $p_{boostrap}$ = 0.018; DG/CA3: t(15) = 3.83,

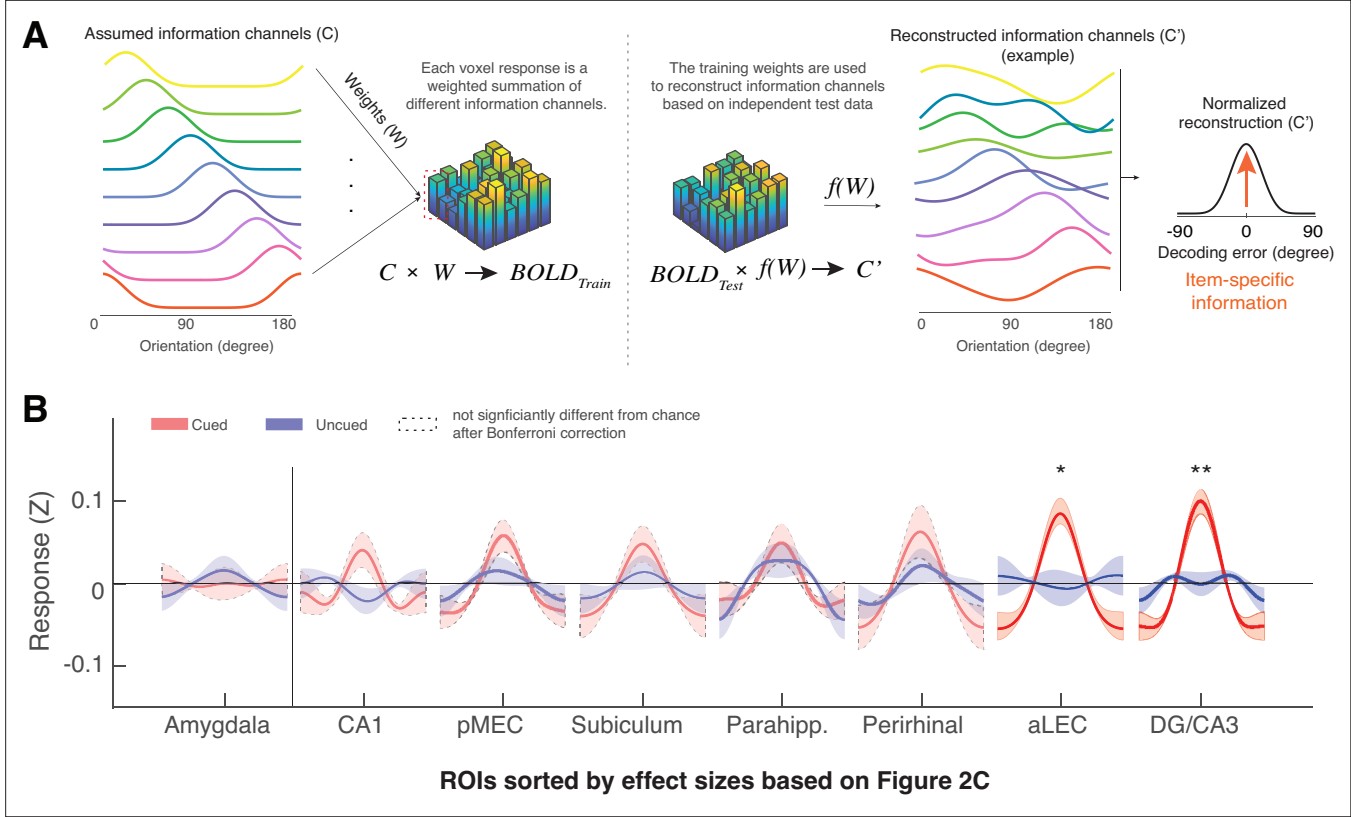

**Figure 3.** The MTL retains item-specific WM information revealed by Inverted Encoding Modeling (IEM). (**A**) The IEM method assumes that each voxel response in the multi-voxel pattern reflects a weighted summation of different ideal stimulus information channels (**C**). The weights (**W**) of these information channels are learned from training data and then applied to independent hold-out test data to reconstruct information channels (**C'**). After shifting these reconstructed information channels to a common center, the resultant vector length of this normalized channel response reflects the amount of retained information on average (also see *Figure 3—figure supplement 1*). (**B**) We find that the BOLD signals from both the aLEC and DG/CA3 contain a significant amount of item-specific information for the cued item, relative to the uncued item. Shaded areas represent the standard error of the mean (s.e.m.) across participants. To retain consistency, we sorted the *x*-axis (ROIs) based on *Figure 2C*. *p<0.05 and **p<0.01 for the comparison of the results based on cued versus uncued items; a.u.=arbitrary unit; aLEC = anterior-lateral entorhinal cortex; pMEC = posterior-medial entorhinal cortex; parahipp. = parahippocampus. Results from detailed statistical tests are summarized in *Supplementary file 1b*.

The online version of this article includes the following figure supplement(s) for figure 3:

**Figure supplement 1.** Example channel responses before and after shifting to the cued orientation for aLEC (**A**) and the amygdala (**B**).

**Figure supplement 2.** Distributed brain regions retain information about the cued item during WM.

**Figure supplement 3.** Stimulus-based representational similarity analysis (RSA) and inverted encoding model (IEM) reveal shared item-related variance in the observed neural data.

**Figure supplement 4.** Analyses based on the whole hippocampus, as compared with a benchmark sphere ROI in the posterior parietal cortex (e.g., superior parietal lobule, SPL) and the hippocampal DG/CA3 subfield.

**Figure supplement 5.** Time-varying IEM analysis shows that mid-delay period activity in aLEC-DG/CA3 contains item-specific information that could not be attributed to perceptual processing alone.

**Figure supplement 6.** Modeling results of the hippocampal subfields based on FreeSurfer labels.

p=0.0016, Cohen's d=0.99, $p_{boostrap}$ = 0.0023). Critically, the amount of information specific to the cued item in the aLEC and DG/CA3 was significantly greater than that in the amygdala control ROI, which is supported by a significant cue (cued vs. uncued) by region (combined aLEC-DG/CA3 vs. amygdala) interaction effect on IEM reconstruction outcomes (F(1, 15)=7.16, p=0.016, $p_{boostrap}$ = 0.010).

Collectively, results from complementary analytical procedures suggest that the MTL's entorhinal-DG/CA3 pathway retains precise item-specific WM content for a simple surface feature (e.g. orientation) to allow fine discrimination of different items in the feature space. As such, the stimulus-based prediction of neural similarity is highly correlated with the amount of reconstructed information based on IEM, even though these two analyses are based on different analytical assumptions (e.g. correlation

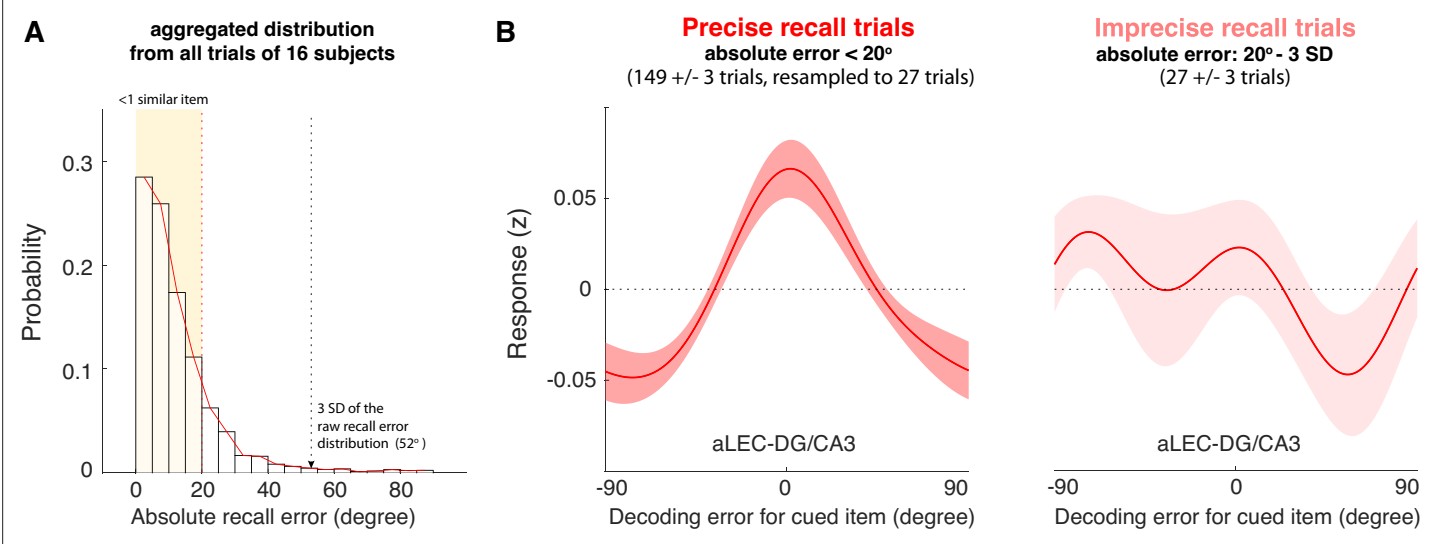

**Figure 4.** The quality of WM information retained in the aLEC-DG/CA3 pathway is associated with later recall fidelity. (**A**) Participants' performance in the visual WM task is high with most of absolute recall errors falling within the 3 SD of the aggregated recall error distribution. As the angular resolution of the presented orientation grating is at least 20° between any two cued items, for most of the trials, participants' recall responses are as precise as within one similar item away from the cued item (i.e. absolute recall error <20°). (**B**) By inspecting the IEM reconstructions for trials with small errors (absolute recall error <20°) and trials with larger errors (absolute recall error: 20° to 3 SD of recall errors), we find that the quality of IEM reconstructions in the combined aLEC-DG/CA3 ROI varies as a function of participants' recall fidelity. Precise recall trials have yielded better IEM reconstruction quality, even after resampling the same number of trials from the data to control for imbalanced trial counts between small- and larger-error trials. Shaded areas represent the standard error of the mean (s.e.m.) across participants.

between IEM and representational similarity analysis for the cued item, aLEC: $r=0.87$, p=0.000012, $p_{boostrap}$ <0.001; DG/CA3: $r=0.78$, p=0.00037, $p_{boostrap}$ <0.001; *Figure 3—figure supplement 3*).

## Reconstruction of WM Item Information in the MTL is associated with recall fidelity

Next, we examined the extent to which WM information retained in the MTL's aLEC-DG/CA3 circuitry is related to an observer's subsequent recall behavior. As the angular resolution of the reconstructed orientation information is 20° in the current study, our data therefore suggest that the MTL can distinguish similar orientation information in WM that is at least 20° apart. This neural separation should be consequential for later recall performance, in that trials with greater item-specific information reconstructed from the MTL should be associated with higher WM recall fidelity. To test this prediction, we grouped the trials from each participant into two categories. The first category contained small recall error trials, where participants made an effective recall response within one similar item away from the cued item (absolute recall error <20°; 149±3 trials [mean ± s.e.m.]). Another category contained larger recall error trials (27±3 trials) with absolute recall errors that were greater than 20° but smaller than the 3 standard deviations (SD) of the aggregated recall error distribution (*Figure 4A*). These trials would capture participants' imprecise recall responses for the cued item, instead of those with an extra-large recall error that could be attributed to other factors such as attentional lapses (*deBettencourt et al., 2019*). The two identified categories of trials together account for about 98% of the total trials (i.e. 176 out of 180 trials).

We then performed the leave-one-block-out analysis to obtain trial-by-trial IEM reconstructions based on delay-period BOLD signals aggregated from the aLEC and DG/CA3. We averaged the IEM reconstructions from the small- and larger-error trials separately. Because trial counts between categories were not balanced, we resampled the data from the small-error trials based on the number of larger-error trials for 5000 times. We took the average of IEM reconstruction across iterations to obtain robust subject-level trial-average estimates with a balanced trial count across different behavioral trial types (*Xie et al., 2020a*; *Yaffe et al., 2014*). By contrasting these estimates at the subject level, we found that the small-error trials yielded significant IEM reconstructions for the cued item

(t(15) = 4.50, p=4.21e-04, Cohen's d=1.16, $p_{boostrap}$ <0.001), whereas the larger-error trials did not (t(15) = 0.03, p=0.98, Cohen's d=0.007, $p_{boostrap}$ = 0.90; *Figure 4B*). Furthermore, the reconstructed WM information in the combined aLEC-DG/CA3 showed better quality in the small-error trials, as compared with that in the larger-error trials (t(15) = 2.45, p=0.027, Cohen's d=0.61, $p_{boostrap}$ = 0.032).

In addition to using an empirical criterion to separate in-memory trials from those extra-large error trials susceptible to occasional attentional lapses (*deBettencourt et al., 2019*), we have also tried another thresholding heuristic. As shown in *Figure 1A*, most trials from each participant fall within this 45° of absolute recall error (i.e. half of the 90° range), and the trials larger than this number are rare (~5 out of 180 trials). We, therefore, used 45° of absolute recall error as a cut-off to identify the imprecise recall trials that were greater than 20° but smaller than 45° of absolute recall error. We performed the same analysis to obtain trial-by-trial IEM reconstructions based on delay-period BOLD signals aggregated from the aLEC and DG/CA3 as outlined above, and then resampled the same number of trials to estimate the IEM reconstructions for the small-error and larger-error trials (<20° vs. 20° - 45° of absolute recall error). Consistent with the 3-SD heuristic, we found that the small-error trials identified by the 45° cut-off heuristic also yielded significant IEM reconstructions for the cued item (t(15) = 4.34, p=5.74e-04, Cohen's d=1.12, $p_{boostrap}$ <0.001), whereas the larger-error trials did not (t(15) = –0.69, p=0.50, Cohen's d=–0.18, $p_{boostrap}$ = 0.67). We then contrasted the difference in IEM reconstructions between these small- and large-error trials across participants. We found that IEM reconstruction for the cued item from the combined aLEC-DG/CA3 showed better quality in the small-error trials, as compared with that in the larger-error trials (t(15) = 3.41, p=0.004, Cohen's d=0.88, $p_{boostrap}$ = 0.008). Collectively, these results suggest that higher-quality WM representation in the entorhinal-DG/CA3 pathway during the delay period is associated with better subsequent recall fidelity and that this association is robust to the selection of cut-off scores for extra-large recall errors.

## Discussion

Based on high-resolution fMRI, this current study uncovers an often-neglected role of the MTL's the entorhinal-DG/CA3 pathway in item-specific WM representation at a minimal task load. Our data suggest that the entorhinal-DG/CA3 circuitry retains item-specific information to allow fine discrimination of similar WM items across trials. The quality of item-specific WM information in the entorhinal-DG/CA3 pathway is associated with an observer's subsequent recall fidelity. Together, these findings fill a missing link in the growing literature regarding the contribution of the MTL to item-level WM representation with a lower information load (*Johnson et al., 2018*; *Sreenivasan and D'Esposito, 2019*).

Theoretically, our findings are consistent with recent neural theories that highlight the involvement of distributed brain areas for WM (*Christophel et al., 2017*; *Eriksson et al., 2015*; *Sreenivasan and D'Esposito, 2019*), including mechanisms in the MTL that are traditionally deemed irrelevant for human WM (*Beukers et al., 2021*; *Borders et al., 2022*; *Goodrich et al., 2019*; *Goodrich and Yonelinas, 2016*). Our findings are built upon the established literature on the entorhinal-DG/CA3 circuitry and the formation of high-fidelity long-term episodic memory (*Aimone et al., 2011*; *Bakker et al., 2008*; *Ekstrom and Yonelinas, 2020*; *Korkki et al., 2021*; *Leal and Yassa, 2018*; *Marr, 1971*; *Reagh and Yassa, 2014*; *Yassa and Stark, 2011*). This function has been linked with various neuronal properties along the entorhinal-DG/CA3 pathway – such as abundant granule cells, strong inhibitory interneurons, and powerful mossy fiber synapses – which could enable sparse coding of information to minimize mnemonic interference (*Aimone et al., 2011*; *Rolls, 2016*; *Rolls, 2013*; *Sahay et al., 2011*). As such, similar information can be retained with a sufficient representational distance to support behavioral discrimination (*Bakker et al., 2008*; *Burke et al., 2011*; *Hunsaker et al., 2008*; *Leal and Yassa, 2018*; *Leutgeb et al., 2007*; *Lohnas et al., 2018*; *Montchal et al., 2019*; *Reagh and Yassa, 2014*; *Ryan et al., 2012*; *Sakon and Suzuki, 2019*). Our data suggest that the same MTL mechanism can also be used to support the quality of WM representation (*Xie et al., 2020b*). Conceptually, potential interference between items either across or within trials would place a demand on pattern separation even over a short delay (*Oberauer and Lin, 2017*). As such, the MTL circuitry involved in the resolution of mnemonic interference (*Aimone et al., 2011*) would play a key role in reducing inference between WM content and other similar information in the feature space. Our data suggest that this process would result in more similar and stable representations for the same remembered item across trials, as detected by multivariate correlational and decoding analyses. However, under certain

task conditions (e.g. learning spatial routes in a naturalistic task over many repetitions), the MTL may maximally orthogonalize overlapping information to opposite representational patterns (hence 'repulsion') to minimize mnemonic interference (*Chanales et al., 2017*). It remains to be determined how these learning-related mechanisms in a more complex setting are related to MTL's contributions to WM representation of simple stimulus features.

Empirically, our results have resolved an issue concerning the decodability of item-specific WM content in the MTL for simple stimulus features. Previously, MTL activity has been shown to scale with WM set size of letters and color squares without decodable item-specific WM content (*Boran et al., 2022*; *Boran et al., 2019*). When item information is shown, it often involves complex stimuli with rich information content (e.g. *Kamiński et al., 2017*). These observations raise the conceptual question concerning the extent to which the MTL responds to task difficulty or retains WM content. In other words, is the MTL not sensitive to simple stimuli or lower task demands at all? Here, with improved spatial resolution of MTL recordings and using a simple stimulus feature, our data suggest that the MTL retains item-level WM information even when the effective WM set size is one. The lack of significant observations in some previous studies using the same paradigm may be due to the lack of granularity in MTL recordings (e.g. *Ester et al., 2015*). To test this, we aggregated data from all the voxels in the hippocampus to examine whether blurred MTL signals would be sufficient to reveal item-specific WM content using the current IEM procedure. As CA1 and subiculum voxels contain less robust WM information (*Figure 3B*), we predicted that this aggregation procedure would attenuate the evidence for WM information due to the reduction in signal-to-noise ratio. Our data are in line with this prediction (*Figure 3—figure supplement 4*). These results, therefore, highlight the importance of fine-grained MTL signals in revealing item-specific WM content.

Alongside these theoretical and empirical contributions, our data also provide additional insights into the conditions under which the MTL is relevant for WM. First, our findings suggest that the MTL's contribution to WM does not depend on whether task demands exceed a limited WM capacity (*Jeneson and Squire, 2012*), although this account has been proposed when interpreting some recent findings for WM tasks using complex stimuli or a higher memory set size (*Boran et al., 2019*; *Jeneson and Squire, 2012*; *Kamiński et al., 2017*; *Kornblith et al., 2017*; *Libby et al., 2014*). Second, our analysis has focused on the mid-delay activity (*Postle et al., 2000*) and hence our findings could not be explained by the MTL's contribution to WM retrieval (*Shrager et al., 2008*). Furthermore, while our findings do not preclude the potential involvement of the MTL during perceptual encoding (*Bonnen et al., 2021*), perceptual involvement could not account for the results based on the comparison between the cued and uncued items (*Bettencourt and Xu, 2016*; *Ester et al., 2015*; *Harrison and Tong, 2009*). If the MTL primarily contributes to perceptual encoding instead of WM retention, we should have observed a comparable amount of information for both study items in the MTL, as they are presented in the same data acquisition TR before cue onset. Since participants do not know the cued item ahead of time, they need to initially remember both items. In line with this interpretation, a time-varying IEM analysis shows that aLEC-DG/CA3 indeed contains a comparable amount of information related to both the cued and uncued items at an earlier time point in the task (*Figure 3—figure supplement 5*). Yet, during the mid-delay period, aLEC-DG/CA3 contains significant information for the cued relative to the uncued item in a similar way as shown in the previous research (*Ester et al., 2015*; *Harrison and Tong, 2009*). Although it is well acknowledged that the current recording method has its inferential limitations in the time domain, these data unambiguously suggest that the entorhinal-DG/CA3 pathway supports the representation of a retrospectively selected memory item during a short delay – a hallmark of WM (*Lorenc et al., 2021*; *Panichello and Buschman, 2021*).

Several open questions remain to be addressed by future research. First, more data are needed to reveal how WM representation in the MTL is compared with and/or related to that retained in distributed neocortical areas (*Christophel et al., 2017*; *Eriksson et al., 2015*; *Sreenivasan and D'Esposito, 2019*). Although the IEM approach allows the reconstruction of information in neural signals, it is not well-suited to directly compare information reconstruction across brain regions. Such a comparison would be complicated by several issues, including the difference in the number of voxels involved and the lack of interpretability of null results when both brain regions contain some WM information. To improve interpretability, we have used the results based on the uncued item as a within-ROI control and contrasted how information specific to the cued item (cued vs. uncued) differs between MTL ROIs and a theoretically irrelevant control region (i.e. the amygdala). One additional potential approach

is to examine how the representations of remembered items are correlated across brain regions (*Figure 2—figure supplement 2A*). The rationale is that delay-period neural patterns across trials should be correlated for two brain regions containing the same information (*Pillet et al., 2019*), as compared with brain regions that do not hold consistent information (*Figure 2—figure supplement 2B*). We tested this conjecture by examining the neural similarity across trials between the aLEC-DG/CA3 and a benchmark ROI in the superior temporal lobule (SPL) – a region that is consistently linked with item-specific information during visual WM retention both in the current data (*Figure 3—figure supplement 4*) and in the previous research (*Bettencourt and Xu, 2016*; *Ester et al., 2015*; *Xu and Chun, 2006*). Supporting this prediction, we found that the similarity of neural patterns between the aLEC-DG/CA3 and the SPL has increased from the pre-stimulus baseline to the WM retention period (*Figure 2—figure supplement 2C*), which contrasts with the lack of changes in the correlation of across-trial neural patterns between aLEC-DG/CA3 and the amygdala control ROI (*Figure 2—figure supplement 2D*). These data suggest that WM information in the entorhinal-DG/CA3 is similiar to that in a well-recognized neocortical WM-related area (*Bettencourt and Xu, 2016*; *Ester et al., 2015*; *Xu and Chun, 2006*). However, considering the limitation in temporal resolution of the current recording method, it remains unknown how the MTL contributes to the dynamic coding schemes underlying WM maintenance (*Stokes, 2015*). Future research with direct recordings from multiple brain areas would be more suitable to investigate the fine-scale temporal dynamic underlying these similar neural patterns across brain regions during WM.

Second, it remains unknown how the MTL circuitry is tuned to specific stimulus features such as orientations, although one of the analytical tools we used was inspired by findings based on neuronal tuning properties from the visual cortex (*Brouwer and Heeger, 2009*; *Sprague et al., 2018*). This is because the assumed orientation channels in IEM do not reflect the underlying neuronal tuning properties and are interpretable only within the assumed model (*Liu et al., 2018*; *Sprague et al., 2018*). Previous research using this method has therefore primarily focused on inferences related to the presence or absence of information content in the neural data (*Bettencourt and Xu, 2016*; *Brouwer and Heeger, 2009*; *Ester et al., 2015*; *Rademaker et al., 2019*; *Sprague et al., 2016*), instead of properties of neural tuning. In the current study, these IEM results are supported by the less assumption-laden results from stimulus-based representational similarity analysis (*Kriegeskorte and Wei, 2021*). These two approaches are therefore complementary to each other. Nevertheless, these analyses are correlational in nature. Hence, although fine-grained neural representations revealed by these analyses are associated with participants' behavioral outcomes (*Figure 4*), it remains to be determined whether the entorhinal-DG/CA3 pathway contributes to the fidelity of WM representation or also to the process of information selection. Strategies for resolving this issue can involve generalizing the current findings to other WM tasks without an explicit requirement of retrospective information selection (*Xie et al., 2023a*) and/or further exploring how the frontal-parietal mechanisms related to visual selection and attention interact with the MTL system (*Panichello and Buschman, 2021*).

Third, the often-neglected role of the MTL in visual processing needs to be further explored. Our findings suggest that the entorhinal-DG/CA3 pathway in the MTL may play a role in retaining of task-relevant item-specific visual WM content, which could not be attributed to perceptual processing alone. These data adds to a growing body of literature that considers the MTL as an important part of the visual system, serving functions ranging from retinotopic coding (*Knapen, 2021*) to predictive coding (*Hindy et al., 2016*). Although retinotopic coding as a form of perceptual processing could underlie WM representation for orientation information, our data highlight that the MTL is sensitive to the retrospectively selected information – a hallmark of WM (*Lorenc et al., 2021*). Furthermore, while we have used orientation as a simple stimulus feature to minimize long-term memory influences, our results do not preclude the role of this MTL circuitry in remembering other stimulus features, such as colors (*Xie et al., 2023a*). To more precisely reveal the MTL mechanisms that are shared across WM and long-term memory, future research should examine the extent to which MTL voxels evoked by a long-term memory task (e.g., mnemonic similarity task, *Bakker et al., 2008*) can be directly used to directly decode mnemonic content in visual WM tasks using different simple stimulus features.

## Conclusion

In sum, our data demonstrate that the MTL's entorhinal-DG/CA3 pathway retains item-specific WM information, similar to that present in other distributed neocortical areas (*Bettencourt and Xu, 2016*;

*Ester et al., 2015*). These results suggest that neural mechanisms underlying the fidelity of long-term episodic memory (*Aimone et al., 2011*; *Bakker et al., 2008*; *Cappiello et al., 2016*; *Ekstrom and Yonelinas, 2020*; *Korkki et al., 2021*; *Marr, 1971*; *Reagh and Yassa, 2014*; *Yassa and Stark, 2011*) are involved in representing precise item-specific WM content. Our data, therefore, provide broader insights into the fundamental constraints that govern the quality of our memory across timescales (*Xie et al., 2023a*; *Xie et al., 2020b*; *Xie and Zhang, 2017a*; *Xie and Zhang, 2016*).

## Materials and methods

### Participants

Sixteen right-handed participants (mean ± s.e.m.: 21.32 ± 0.73 years old, 8 females) were recruited for the study with monetary compensation ($20/hour). This sample size was designed to be no smaller than that involved in the prior studies using similar experimental paradigms and analytical procedures (*Bettencourt and Xu, 2016*; *Ester et al., 2015*; *Harrison and Tong, 2009*). All participants reported normal or corrected-to-normal visual acuity and no history of neurological/psychiatric disorders or prior psychostimulant use. They provided written informed consent before the study, following the protocol approved by the Internal Review Broad of the University of California, Riverside (reference number: HS-17-035).

### Visual WM task

Participants performed an orientation visual working memory task adapted from previous studies (*Ester et al., 2015*; *Harrison and Tong, 2009*) inside an MRI scanner (*Figure 1A*). Briefly, on each trial, we sequentially presented two sine-wave gratings (~4.5° of visual angles in radius, contrast at 80%, spatial frequency at ~1 cycle per visual degree, randomized phase) at the center of the screen. Each grating appeared for 200ms, with a 400 ms blank screen in between. The two gratings had different orientations randomly drawn from nine predefined orientations (0–160° in 20° increments) and were >20° away from one another (see *Figure 1—figure supplement 1*). They were presented with a small random angular jitter (±1° to 5°). Following the offset of the second grating of each pair by 400ms, we presented a cue ('1' or '2', corresponding to the first or second grating, respectively) for 550ms to indicate which grating orientation the participant should remember and maintain over an 8750 ms delay period. We instructed participants to remember only the cued grating and to ignore the uncued one. After the delay period, we presented a test grating initially aligned to a random orientation. Participants then pressed the response box buttons to continuously adjust the test grating until it matched the orientation of the cued grating based on their memory. We asked the participants to make a response within 3500ms following the onset of the test grating (averaged median response time across participants: 2929±156ms). After the response, we provided feedback to the participants by presenting a line marking the correct orientation, which was followed by an inter-trial interval of 3500 or 5250ms. Participants completed 10 blocks of 18 trials, yielding a total of 180 trials inside the scanner. Before scanning, they completed 2 blocks of 18 trials outside the scanner for practice. The cue position and the orientations of presented gratings were randomly intermixed within each block.

Under an effective set size of one item, participants' recall performance was high (*Figure 1B*), with most recall errors centered around ±45° of the cued orientation (~97% of the trials) within the ±90° range. Hence, we retained all trials when investigating the amount of WM information in the recorded neural data during the delay period for multivariate analyses. We used the absolute recall error as a trial-level estimate of recall fidelity (*Panichello and Buschman, 2021*), assuming that large recall errors were driven by imprecise WM instead of other factors, such as occasional attentional lapses (*deBettencourt et al., 2019*; *Xie and Zhang, 2017b*). To minimize the contamination of these factors in linking the neural data with the behavioral data, we would focus on the trials where participants have recalled within the 3 SD of the aggregated recall error distribution (*Figure 4A*; see details in a subsequent section).

### MRI data acquisition and pre-processing

We acquired neuroimaging data using a 32-channel sensitivity encoding (SENSE) coil in a Siemens Prisma 3.0-Tesla scanner. We first acquired a high-resolution 3D magnetization-prepared rapid gradient echo (MP-RAGE) structural scan (0.80 mm isotropic voxels) and then functional MRI scans

consisted of a T2*-weighted echo-planar imaging (EPI) sequence: TR = 1750ms, TE = 32ms, flip angle = 74°, 69 slices, 189 dynamics per run, 1.5×1.5 mm² in-plane resolution with 2 mm slice thickness, FOV read = 222 mm, FOV phase = 86.5%. This sequence was optimized for high-resolution functional MRI with whole-brain coverage for the scanner. Each functional run lasted 5 min and 30.75 s. At the end of the experiment, we acquired two additional scans with opposite phases to correct for EPI distortions (*Irfanoglu et al., 2015*).

We preprocessed neuroimaging data using the *Analysis of Functional NeuroImages* (AFNI) software (*Cox, 1996*). Briefly, functional data were de-spiked (*3dDespiked*), slice timing corrected (*3dtshift*), reverse-blip registered (*blip*), aligned to structural scan (*align_epi_anat.py*), motion-corrected (*3dvolreg*), and masked to exclude voxels outside the brain (*3dautomask*). To avoid introducing artificial autocorrelations in later analyses, functional data were not smoothed. For the same reason, we extracted the raw BOLD signals from the middle 3 TRs of the 5-TR retention interval for later analyses without fitting the data to the hemodynamic model (*Ester et al., 2015*). These raw BOLD signals were z-scored within each block/run, before extracting the TRs of interest. In particular, we convolved the data from the 5 TR delay period with a set of weights (i.e. 0, 1, 2, 1, 0) that resembled the TENT function in AFNI to maximize the inclusion of mid-delay activity for later analysis (*Postle et al., 2000*). This approach factors in 5–6 s of hemodynamic adjustment (*Lewis-Peacock and Postle, 2008*) and has been considered fundamentally conservative in estimating delay-period activity (*Feredoes and Postle, 2007*). This approach also provides a reasonable estimate for the BOLD response around a given TR with an improved signal-to-noise ratio without assuming the shape of the underlying hemodynamic response (*Chen et al., 2015*). We also performed the time-varying version of this analysis by shifting the peak of the TENT function over time (see *Figure 3—figure supplement 5* for details).

To retain the consistency with the prior research, we defined participant-specific MTL ROIs (bilateral hippocampal DG/CA3, CA1, and subiculum, entorhinal/perirhinal cortex, and parahippocampus, see *Figure 2A*) based on the T1 image using the same segmentation algorithm from the previous studies (*Montchal et al., 2019*; *Reagh et al., 2017*). In brief, using the *Advanced Normalization Tools* (*Avants et al., 2008*), this algorithm aligned an in-house segmented template to each participant's T1 image. This template contains manually labeled ROIs for hippocampal subfields (DG, CA3, CA1, subiculum) and other verified MTL subregions (aLEC, pMEC, perirhinal, and parahippocampus). The efforts to select and verify these MTL ROIs have been detailed in previous studies (*Montchal et al., 2019*; *Reagh et al., 2018*). In brief, in addition to the commonly identified perirhinal and parahippocampus ROIs, hippocampal subfields were manually identified and aggregated from a set of T1 and T2 atlas images based on prior harmonized efforts (*Yushkevich et al., 2015*). Entorhinal ROIs (aLEC and pMEC) were added to the template from a previous study (*Maass et al., 2015*). For functional analysis, we combined DG and CA3 subfields as a single label given the uncertainty in separating signals from them in fMRI data (*Reagh et al., 2017*). In addition, we also verified our findings in hippocampal subfields based on a different segmentation protocol via *FreeSurfer* (*Iglesias et al., 2015*), which yielded consistent findings (*Figure 3—figure supplement 6*). Therefore, our current observations are unlikely to be limited to a specific parcellation procedure of hippocampal subfields.

Furthermore, we identified subject-specific segmented amygdala as a control ROI based on participant-specific *Freesurfer* parcellation (*Saygin et al., 2017*). The amygdala is a part of the limbic system traditionally considered a central brain region processing emotion-laden information. Because the task stimuli (orientation gratings) and testing procedure (no reward manipulation) in the current study are emotionally neutral, the amygdala is therefore theoretically irrelevant for the current study (*Iwai et al., 1990*; *Xie et al., 2022*; *Xie and Zhang, 2016*). Furthermore, as its signal-to-noise ratio is similar to adjacent structures, the amygdala can serve as a control site for the observation in other MTL ROIs.

## Stimulus-based representational similarity analysis

To examine whether MTL delay-period activity can distinguish different cued orientation gratings, we performed a stimulus-based representational similarity analysis (*Kriegeskorte and Diedrichsen, 2019*). The rationale is that if the recorded neural data contain information to allow fine discrimination of the cue item, the neural data should track the feature distance between any pair of cued items across trials to allow fine discrimination of these items (*Kriegeskorte and Wei, 2021*). Hence, we first calculated the stimulus similarity pattern across trials using 180 minus the absolute angular distance

between the orientation labels of every two trials (*Figure 2B*, top panel). Next, we calculated the cosine similarity of the delay-period neural signals *B* across n voxels from the middle 3 TRs in every pair of trials (*Figure 2B*, bottom panel). This yields a trial-by-trial matrix in which the similarity between voxel response vectors $B_i$ and $B_j$ can be calculated as,

$$S\left(B_i, B_j\right) = \frac{B_i \cdot B_j}{\|B_i\| \, \|B_j\|}$$

Finally, we correlated the neural similarity pattern and stimulus similarity pattern across trials (rank-order and Fisher's transformed, *Xie et al., 2018*) to gauge how the recorded neural signals track the stimulus features across trials.

## Inverted encoding modeling (IEM)

To decode item-level information from the raw BOLD signals (*Ester et al., 2015*), we first constructed a linear encoding model to represent orientation-selective responses in multi-voxels of activity from a given brain region. We did not impose any additional feature selection procedures other than using the anatomically defined ROIs to identify relevant multi-voxel features in this analysis (see *Supplementary file 1c* for the number of voxels/features included for each subject in each ROI). We assumed that the response of each voxel is a linear summation of 9 idealized information channels (*Figure 2B*), estimated by a set of half-wave rectified sinusoids centered at different orientations based on the tuning profile of orientation-sensitive neural populations. Hence, we formalized the observed raw BOLD signals *B* (m voxels ×n trials) as a weighted summation of channel responses *C* (k channels ×n trials), based on the weight matrix, *W* (m voxels ×k channels), plus residual noise (*N*),

$$B = WC + N$$

Given $B_1$ and $C_1$ from a set of training data, the weight matrix can be calculated as,

$$W = B_1 C_1^T \left(C_1 C_1^T\right)^{-1}$$

The training weight matrix *W* was used to calculate a set of optimal orientation filters *V*, to capture the underlying channel responses while accounting for correlated variability between voxels (i.e. the noise covariance), as follows,

$$V_i = \frac{\sum_i^{-1} W_i}{W_i^T \sum_i^{-1} W_i}$$

where $\Sigma_i^{-1}$ is the regularized noise covariance matrix for channel i (1–9), estimated as,

$$\Sigma_i^{-1} = \frac{1}{n_1 - 1} \varepsilon_i \varepsilon_i^T$$

$$\varepsilon_i = B_1 - W_i C_{1, i}$$

Here, $n_1$ is the number of training trials, and $\varepsilon_i$ is a matrix residual based on the training set $B_1$ and is obtained by regularization-based shrinkage using an analytically determined shrinkage parameter. Next, for the independent hold-out test dataset $B_2$, trial-by-trial channel responses $C_2$ are calculated as follows,

$$C_2 = V^T B_2$$

We used a leave-one-block-out cross-validation routine to obtain reliable estimate channel responses for all trials. For each participant, in every iteration, we treated all but one block as $B_1$ and the remaining block as $B_2$ for the estimation of $C_2$. This analysis yielded estimated channel responses $C_2$ for each trial, which were interpolated to 180° and circularly shifted to a common center (0°, by convention). We reconstructed these normalized channel responses separately using orientation labels of the cued item, the uncued item, and shuffled orientations. We then quantified the amount of item-related information (R) by converting the average channel response (z) to polar form given $\psi$

as the vector of angles at which the channels peak ($z = Ce^{2i\psi}$). We then projected them onto a vector with an angle of 0°,

$$R = |z| \cos\left(arg(z)\right)$$

With whole-brain coverage, we performed an additional searchlight procedure in combination with the IEM analysis to replicate the previous findings (*Ester et al., 2015*). First, we normalized participants' brain data to an MNI template using the *Advanced Normalization Tools*. Second, we defined a spherical 'neighborhood' (radius 8.0 mm) centered on voxels in a cortical mask containing only gray matter voxels. We discarded neighborhoods with fewer than 100 voxels. Last, we estimated item-related information (*R*) about the to-be-remember item based on the IEM analysis outlined above to assess WM information within each searchlight sphere. We obtained consistent findings as compared with the previous findings (*Figure 3—figure supplement 2*), suggesting the reliability of the current data.

## Linking IEM reconstruction with behavioral recall performance

To examine how the IEM reconstruction of the cued item in the aLEC-DG/CA3 pathway is associated with recall fidelity, we performed the IEM analysis based on data combined from the aLEC and DG/CA3 ROIs. Similar to the analytical framework outlined above, we split each participant's data into random blocks of 18 trials and then perform a leave-one-block-out analysis to obtain IEM reconstructions for all trials in each block based on the weights trained from other blocks. As this analysis is agnostic to participants' recall performance at this stage, if IEM reconstruction is not associated with participants' recall fidelity, the reconstructed information channels should be comparable regardless of recall errors. To test against this prediction, we split participants' data into small- and larger-error trials. First, as the angular resolution was at least 20° for any two items in the current design, we defined small-recall error trials as those in which participants had reported within one similar item away (absolute recall error <20°; 149±3 trials). Next, to separate larger-recall errors based on less precise WM representation from those attributable to attention lapses (*deBettencourt et al., 2019*), we adopted a widely-used thresholding heuristic to find potentially different categories of data points based on the empirical SD of a distribution. Specifically, in our current data, we first calculated the empirical SD (17.33°) of the aggregated raw recall error distribution from all subjects across 2880 trials (ranging from –90° to 90°), which captures the overall variability in participants' recall performance without a priori model assumption. We then retained the larger-recall error trials within 20° to 3 SD of the recall error distribution (27±3 trials; *Figure 4A*). These larger-error trials presumably contain mostly imprecise recall responses, instead of infrequent extra-large errors that could be attributed to other factors like attentional lapses (*deBettencourt et al., 2019*). Considering that most of the trials have a recall error of ±45° out of the ±90° range in every subject by visual inspection (97% of the trials, *Figure 1B*), we have also used 45° of absolute recall error as a cut-off for extra-large error trials and obtained similar findings in subsequent analyses.

To balance the trial counts between these two categories of trials, we resampled the same number of trials based on the number of larger-error trials from the small-error trials for 5000 times. This resampling procedure has ensured that the average IEM reconstruction from the small-error trials was estimated based on the same number of trials as compared with the larger-error trials – an approach often used to obtain less biased estimates of neural measures across different behavioral trial types (*Xie et al., 2020a*; *Yaffe et al., 2014*). We contrasted the difference in IEM reconstructions for the cued item in the aLEC-DG/CA3 between these two categories of trials across participants.

## Statistical rocedures

We evaluated statistical significance based on conventional within-subject statistical procedures, such as paired-sample t-tests, with two-tailed p values. Similar results were obtained and verified based on non-parametric statistics (e.g. bootstrapped p values) that have few analytical assumptions (*Good, 2013*). In particular, we resampled participants' data with replacement over 1000 iterations and calculated the empirical two-tailed p values (note: these p values can slightly vary across different iterations of resampling and those smaller than 0.001 are marked as $p_{bootstrap}$ <0.001). We estimated the size of these effects based on Cohen's *d*. Except for pre-defined contrast analysis (e.g. cued vs. uncued), we corrected for multiple comparisons by using Bonferroni correction with an alpha level set as 0.05

(*Rosenthal and Rubin, 1983*). For visualization of variability in mean estimates, we have used the standard error of the mean across participants (s.e.m.), namely the standard deviation of a measure divided by the square root of sample size, as error bars (or areas) in *Figures 2–4*.

## Acknowledgements

We thank all the participants who selflessly volunteered their time for this study. We thank Nicholas J Tustison and Jason Langley for their technical support. This work was supported by the National Institute of Mental Health (1R01MH117132, PI: W Z). WX was funded by the National Institute of Neurological Disorders and Stroke Competitive Postdoctoral Fellowship Award and the NIH Pathway to Independence Award (K99NS126492).

## Additional information

### Funding

| Funder | Grant reference number | Author |
| --- | --- | --- |
| National Institute of Mental Health | R01MH117132 | Weiwei Zhang |
| National Institute of Neurological Disorders and Stroke | ZIA-NS003144 | Kareem A Zaghloul |
| National Institute of Neurological Disorders and Stroke | NCFA | Weizhen Xie |
| National Institute of Neurological Disorders and Stroke | K99NS126492 | Weizhen Xie |

The funders had no role in study design, data collection and interpretation, or the decision to submit the work for publication.

### Author contributions

Weizhen Xie, Weiwei Zhang, Conceptualization, Resources, Data curation, Software, Formal analysis, Supervision, Funding acquisition, Validation, Investigation, Visualization, Methodology, Writing – original draft, Project administration, Writing – review and editing; Marcus Cappiello, Data curation, Investigation, Writing – original draft, Project administration; Michael A Yassa, Methodology; Edward Ester, Software, Formal analysis, Validation, Investigation, Visualization, Methodology; Kareem A Zaghloul, Resources, Formal analysis, Investigation, Writing – original draft, Writing – review and editing

### Author ORCIDs

Weizhen Xie http://orcid.org/0000-0003-4655-6496
Michael A Yassa http://orcid.org/0000-0002-8635-1498
Kareem A Zaghloul http://orcid.org/0000-0001-8575-3578

### Ethics

Human subjects: Participants provided written informed consent before the study, following the protocol approved by the Internal Review Broad of the University of California, Riverside.

### Decision letter and Author response

Decision letter https://doi.org/10.7554/eLife.83365.sa1
Author response https://doi.org/10.7554/eLife.83365.sa2

## Additional files

### Supplementary files

• Supplementary file 1. Supporting data tables. (A) Tests of statistical significance in the neural

similarity across trials captured by the similarity of the cued item. (B) Tests of statistical significance in IEM results for the cued item. (C) The number of voxels included for each bilateral ROI in each subject.

- MDAR checklist

### Data availability

Non-identified data (e.g., MTL activities across ROIs and trial-by-trial behavior responses) and custom codes are available via the Open Science Framework repository (https://osf.io/zvdnr/).

The following dataset was generated:

| Author(s) | Year | Dataset title | Dataset URL | Database and Identifier |
| --- | --- | --- | --- | --- |
| Xie W, Cappiello M, Yassa MA, Ester E, Zaghloul K, Zhang W | 2023 | The Entorhinal-DG/CA3 Pathway in the Medial Temporal Lobe Retains Visual Working Memory of a Simple Surface Feature | https://osf.io/zvdnr/ | Open Science Framework, zvdnr |

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
