## [Editor Report]

This useful study highlights the contribution of the medial temporal lobe (MTL), and the DG/CA3 hippocampal pathway in particular, to neural activity during the working memory delay period. The evidence supporting this is compelling, using diverse state-of-the-art approaches to neural data analysis and relating it to behavioural data. The work will be of significant interest to neuroscientists specialising in the research area of human working memory.

---

## [Decision Letter]

**Decision letter after peer review:**

Thank you for submitting your article "The Entorhinal-DG/CA3 Pathway in the Medial Temporal Lobe Retains Visual Working Memory of a Simple Surface Feature" for consideration by *eLife*. Your article has been reviewed by 3 peer reviewers, and the evaluation has been overseen by a Reviewing Editor and Floris de Lange as the Senior Editor. The reviewers have opted to remain anonymous.

Essential revisions:

1) The reviewers raised several issues of interpretation:

a) It is not clear whether the authors propose that the entorhinal-DG/CA3 circuitry retains specific visual feature (i.e., orientation and not other features) or enables discrimination among similar items (item1 vs. item2 independent of maintained feature) (see R1).

b) It is unclear whether the more distinct representation in MTL for the cued item may reflect hippocampal repulseion (see R2).

The authors should discuss these issues. We would welcome it if the authors could corroborate their interpretation with new data/analysis, but otherwise it would be important that the interpretational limitations are acknowledged in the manuscript.

2) The application of IEM methods should be motivated more clearly in the manuscript (see R2).

3) The behavioral data should be analyzed in greater detail (see R3).

Please see below for a full description of issues raised by the reviewers.

*Reviewer #1 (Recommendations for the authors):*

The study by Xie et al., investigates whether the entorhinal-DG/CA3 pathway is involved in working memory maintenance. The study consists of 16 participants who perform an orientation visual working memory task with a retro-cue. Briefly, participants are presented with two gratings each with a different orientation followed by a cue that indicates which orientation should be further maintained. The authors focused their analyses on the delay interval following the cue. The main findings include a correlation between stimulus and neural similarities that was specific for cued stimulus and entorhinal-DG/CA3 locations. The authors observed similar results (cuing and region specificity) using inverted encoding modeling approach. Finally, they also showed that trials in which participants made a smaller error showed a better reconstruction fidelity on the cued side (compared to un-cued). This effect was absent for larger-error trials.

The study challenges a widely held traditional view that working memory and episodic memory have largely independent neural implementations. Specifically, this traditional view posits that the MTL is critical for episodic memory but not for working memory. The study adds to a large body of evidence showing involvement of the hippocampus across a range of different working memory tasks and stimuli. Nevertheless, it still remains unclear what functions may the hippocampus play in working memory.

1) The authors interpret their findings as suggesting that the entorhinal-DG/CA3 circuitry retains item-specific information to allow fine discrimination of similar WM items across trials. It was not clear to me whether they propose that the entorhinal-DG/CA3 circuitry retains specific visual feature (i.e., orientation and not other features) or enables discrimination among similar items (item1 vs. item2 independent of maintained feature). One other alternative interpretation of their results is that the entorhinal-DG/CA3 circuitry participates in orienting attention to memory representation or otherwise supports selection between initially remembered orientations. The current approach cannot tell these alternatives apart which is the main limitation of the study.

2) If the authors propose that the hippocampus maintains item-specific information about orientation (p. 9), it would be important to show specificity and also discuss how hippocampal contribution to working memory differs from the contribution of the visual system (e.g., Harrison and Tong, 2009)? On the other hand if authors propose that hippocampus enables discriminating among similar items independent of maintained feature, it would be important to show whether this generalize to other features.

3) Furthermore, different features have been shown to elicit sustained or rhythmic neural activity in the hippocampus. To corroborate the functional interpretation put forth by the authors, it would thus be important to relate the current results to what is already known about hippocampal mechanisms of working memory maintenance.

*Reviewer #2 (Recommendations for the authors):*

1. It seems that a traditional way to quantify the performance in the retro-cue WM task is to analyze the reaction times. I wonder if the authors analyzed it along with the recall error and if it could be informative.

2. I would like to make sure if all the presented ROI results were Bonferroni-corrected for multiple comparisons, i.e. if the original α-value was divided by the number of analyses on the dependent variable, here the number of ROIs.

3. I was wondering if the authors could explain why they selected bilateral ROIs instead of more specific left and right hemisphere regions. Could there be some laterialization effects? Do I understand correctly that the ROI analyses were performed on an average signal from the left and right hemisphere?

4. Could the authors clarify if the representational similarity analysis and IEM were performed on the cued vs. uncued items, regardless of the recall error? The authors performed a nice follow-up control analysis of the IEM reconstructions across precise recall and imprecise recall trails, how about the representational similarity analysis?

5. Related to the control analysis of the IEM reconstructions across precise recall and imprecise recall trails – could the authors explain why did they combine data from aLEC and DG/CA3 for this analysis, if in all the other analyses they analyse these ROIs separately?

6. In the caption of Figure 2B the authors mention only correlating the feature similarity of every two cued items with the neural pattern similarity during the WM delay period, but in the analysis (described in C) the effect sizes were compared with the uncued items. Thus, the authors could mention analyzing the uncued items also in B, for the clarity.

*Reviewer #3 (Recommendations for the authors):*

Recommendations for improving the writing and presentation

– Check for consistent use of past tense (especially prevalent in Methods)

– Add a more detailed explanation of what model is fitted on which data during the knee-point thresholding, given that part of the results hinge on separation of high- and low-error trials and therefore the exclusion of faulty trials.

– Consider including this review in the introduction, Christophel, T. B., Klink, P. C., Spitzer, B., Roelfsema, P. R., and Haynes, J. D. (2017). The distributed nature of working memory. Trends in cognitive sciences, 21(2), 111-124.

– P.6 mid-page, note that significance and effect size are not the same.

– Consider putting paragraph 2 last since 2 is very speculative. Please make sure your wording does not suggest that your data support the role of MTL in reducing interference in WM.

– In the discussion consider picking up on concepts from the introduction on why you think you found this effect with simple stimuli whereas other studies claim MTL involvement only for more complex stimuli or high-demand tasks.

[Editors' note: further revisions were suggested prior to acceptance, as described below.]

Thank you for resubmitting your work entitled "The Entorhinal-DG/CA3 Pathway in the Medial Temporal Lobe Retains Visual Working Memory of a Simple Surface Feature" for further consideration by *eLife*. Your revised article has been evaluated by Floris de Lange (Senior Editor) and a Reviewing Editor.

All reviewers were happy with your revisions and recommended that the manuscript be accepted for publication. Reviewer #1 however had some small final comments with respect to the statistical analyses (suggesting to only use non-parametric tests when assumptions for parametric tests are not met; or skip the parametric tests altogether and only report non-parametric tests).

You will find the individual reviewer reports below.

*Reviewer #1 (Recommendations for the authors):*

The authors responded to most of my comments. Some concerns arise as a result of the revisions. Neither in the manuscript nor in the response the authors specified how they ensured assumptions of statistical tests are met. I do appreciate that their results are consistent when using non-parametric alternatives but it would be important to report these results in the manuscript.

I suggest to either explicitly test assumptions and wherever those are not met using non-parametric alternatives or skip the parametric statistics and use non-parametric tests to start with. Otherwise, the authors risk reporting tests that are not valid, particularly given their relatively small sample size.

*Reviewer #2 (Recommendations for the authors):*

I have now had the chance to read the revised manuscript and author responses. All my concerns, as well as the concerns of other reviewers, were addressed. The authors now included several clarifications in the methods and results sections, clarifications that facilitate a general understanding of the experiment and help future attempts to replicate the findings. Moreover, the authors extended the discussion section by commenting on interesting alternative interpretations of the results. Overall, the authors did an excellent job revising the manuscript and in the present form it presents compelling evidence for significant findings.

*Reviewer #3 (Recommendations for the authors):*

Thank you to the authors for their resubmission and responses to my comments.

Generally, the revised manuscript thoroughly addresses the comments I have made. The weaknesses I outlined have been acknowledged in the discussion. All of my recommendations have been followed, either by clarification in the text or addition of a figure.

Overall, the responses are convincing and their resubmission is satisfactory.

---

## [Author Response]

Essential revisions:1) The reviewers raised several issues of interpretation:a) It is not clear whether the authors propose that the entorhinal-DG/CA3 circuitry retains specific visual feature (i.e., orientation and not other features) or enables discrimination among similar items (item1 vs. item2 independent of maintained feature) (see R1)b) It is unclear whether the more distinct representation in MTL for the cued item may reflect hippocampal repulseion (see R2).The authors should discuss these issues. We would welcome it if the authors could corroborate their interpretation with new data/analysis, but otherwise it would be important that the interpretational limitations are acknowledged in the manuscript.

We thank the editor and reviewers for these cautious notes. We have now clarified that our results and interpretation of these results are grounded in the prior research concerning the role of the entorhinal-DG/CA3 circuitry in supporting the quality of long-term episodic memory. Our data highlight that this MTL circuitry can distinguish similar items from the same feature space across trials. As a result of this computation, item-specific information can also be retained in this pathway. Although we have used orientation as a simple stimulus feature to minimize long-term memory influences, our results do not preclude the role of this MTL circuitry in other stimulus features. For instance, we have recently generalized our findings to colors using intracranial EEG and confirmed that trial-by-trial mnemonic discrimination during a short delay is indeed associated with the fidelity of item-specific WM representation (Xie, Chapeton, et al., in press). Please see our response to Reviewer #1 for details on these issues.

Furthermore, we also would like to thank Reviewer #2 for raising an interesting and relevant alternative interpretation concerning hippocampal repulsion. We have now discussed this hypothesis in our revised manuscript. Please see our response to Reviewer #2 for details.

2) The application of IEM methods should be motivated more clearly in the manuscript (see R2).

We thank Reviewer #2 for raising this clarification question. We have addressed this in the response letter and the revised manuscript by highlighting the consistency between this method and other multivariate approaches as well as the empirical efficiency (model-based) in using this method to reveal item-specific information in the neural data. Further details of this method have been extensively discussed and used in the literature (e.g., Ester et al., 2015; Liu et al., 2018; Sprague et al., 2018). Here, we would like to emphasize the benefits of using this method in the current research to be able to compare our findings with that from the previous research (e.g., Ester et al., 2015).

3) The behavioral data should be analyzed in greater detail (see R3).

We thank Reviewer #3 for this note. We have included more details about the behavioral data in a supplementary figure (updated Figure 1—figure supplement 1).

Please see below for a full description of issues raised by the reviewers.Reviewer #1 (Recommendations for the authors):The study by Xie et al., investigates whether the entorhinal-DG/CA3 pathway is involved in working memory maintenance. The study consists of 16 participants who perform an orientation visual working memory task with a retro-cue. Briefly, participants are presented with two gratings each with a different orientation followed by a cue that indicates which orientation should be further maintained. The authors focused their analyses on the delay interval following the cue. The main findings include a correlation between stimulus and neural similarities that was specific for cued stimulus and entorhinal-DG/CA3 locations. The authors observed similar results (cuing and region specificity) using inverted encoding modeling approach. Finally, they also showed that trials in which participants made a smaller error showed a better reconstruction fidelity on the cued side (compared to un-cued). This effect was absent for larger-error trials.The study challenges a widely held traditional view that working memory and episodic memory have largely independent neural implementations. Specifically, this traditional view posits that the MTL is critical for episodic memory but not for working memory. The study adds to a large body of evidence showing involvement of the hippocampus across a range of different working memory tasks and stimuli. Nevertheless, it still remains unclear what functions may the hippocampus play in working memory.

We thank the reviewer’s positive appraisal of the current research, which adds to the growing research interest in the MTL’s contribution to WM.

1) The authors interpret their findings as suggesting that the entorhinal-DG/CA3 circuitry retains item-specific information to allow fine discrimination of similar WM items across trials. It was not clear to me whether they propose that the entorhinal-DG/CA3 circuitry retains specific visual feature (i.e., orientation and not other features) or enables discrimination among similar items (item1 vs. item2 independent of maintained feature). One other alternative interpretation of their results is that the entorhinal-DG/CA3 circuitry participates in orienting attention to memory representation or otherwise supports selection between initially remembered orientations. The current approach cannot tell these alternatives apart which is the main limitation of the study.

We thank the reviewer for this insightful comment. We have tested whether the MTL keeps track of the feature distance of temporarily remembered items from a simple continuous feature space based on a trial-wise RSA. Our interpretation is that these results indicate the capability of the MTL to distinguish similar items across trials (and hence fine discrimination across items). Conceptually, this could involve the differentiation between items 1 and 2 within a trial, although the current recording method (e.g., slow fMRI signals) does not afford the temporal resolution to reveal this within-trial effect. Grounded in the theoretical relationship between representational similarity and neural coding (Kriegeskorte and Wei, 2021), these trial-wise RSA findings further indicate that there is item-specific information retained in MTL, which is supported by our IEM results. In essence, our current data suggest that the MTL contains information about a retrospectively selected memory item and can tell this item apart from other items in the feature space. Based on these results, the reviewer is correct that we could not distinguish the information retention versus selection processes in the MTL – an inferential limitation that also exists in previous research using similar methods (e.g., Bettencourt and Xu, 2015; Ester et al., 2015; Harrison and Tong, 2009). To resolve these issues, we have recently generalized our findings to colors and confirmed that mnemonic discrimination during a short delay period is indeed associated with the fidelity of item-specific WM representation (Xie et al., in press).

We have discussed these issues in our Discussion.

“… These two approaches are therefore complementary to each other. Nevertheless, these analyses are correlational in nature. Hence, although fine-grained neural representations revealed by these analyses are associated with participants’ behavioral outcomes (Figure 4), it remains to be determined whether the entorhinal-DG/CA3 pathway contributes to the fidelity of the selected WM representation or also to the selection of task-relevant information. Strategies for resolving this issue can involve generalizing the current findings to other WM tasks without an explicit requirement of information selection (e.g., intracranial stimulation of the MTL in a regular WM task without a retro-cue manipulation, Xie et al., in press) and/or further exploring how the frontal-parietal mechanisms related to visual selection and attention interact with the MTL system (Panichello and Buschman, 2021).”

2) If the authors propose that the hippocampus maintains item-specific information about orientation (p. 9), it would be important to show specificity and also discuss how hippocampal contribution to working memory differs from the contribution of the visual system (e.g., Harrison and Tong, 2009)? On the other hand if authors propose that hippocampus enables discriminating among similar items independent of maintained feature, it would be important to show whether this generalize to other features.

We thank the reviewer for this insightful comment. Motivated by the large body of literature on the entorhinal-DG/CA3 circuitry’s function in distinguishing similar information across different stimulus dimensions of episodic long-term memory (e.g., object details, Reagh and Yassa, 2014; temporal information, Montchal et al., 2019), our study is grounded in the hypothesis that the same circuitry may also distinguish similar representations of retained WM items. As a result of this computation, item-specific WM information can be retained along this pathway. These two aspects of findings (i.e., information discrimination and information individualization) are the two sides of the same coin (see some detailed theoretical discussions in Kriegeskorte and Wei, 2021).

We use orientation as a simple task content to minimize rich long-term memory associations often seen in complex task stimuli (e.g., objects and scenes). Our findings therefore should be generalizable to other simple stimulus features, such as colors, as shown in one of our recent studies (Xie et al., in press). Therefore, our findings should not be interpreted that the entorhinal-DG/CA3 circuitry retains only orientation information.

We have clarified this in our revised Discussion.

“Furthermore, while we have used orientation as a simple stimulus feature to minimize long-term memory influences, our results do not preclude the role of this MTL circuitry in remembering other stimulus features, such as colors (e.g., Xie et al., in press). To more precisely reveal the MTL mechanisms that are shared across WM and long-term memory, future research should examine the extent to which MTL voxels evoked by a long-term memory task (e.g., mnemonic similarity task, Bakker et al., 2008) can be directly used to directly decode mnemonic content in visual WM tasks using different simple stimulus features.”

3) Furthermore, different features have been shown to elicit sustained or rhythmic neural activity in the hippocampus. To corroborate the functional interpretation put forth by the authors, it would thus be important to relate the current results to what is already known about hippocampal mechanisms of working memory maintenance.

We thank the reviewer for this critical comment. However, as our findings are not limited to the hippocampus (e.g., also in aLEC) and as our current recording methods preclude inferences based precise timing information, our results cannot directly speak to sustained or rhythmic neural activity observed in the hippocampus during WM retention. Furthermore, as WM maintenance is dynamic, it may depend on multiple forms of neural activity (e.g., sustained, burst, or complex coupling). Hence, more will need to be done before we can link these complex neural dynamics of WM with MTL activity. We have discussed this issue in revised manuscript as a direction for future investigation.

“However, considering the limitation in temporal resolution of the current recording method, it remains unknown how the MTL contributes to the dynamic coding schemes underlying WM maintenance (Stokes, 2015). Future research with direct recordings from multiple brain areas would be more suitable to investigate the fine-scale temporal dynamic underlying these similar neural patterns across brain regions during WM.”

Reviewer #2 (Recommendations for the authors):1. It seems that a traditional way to quantify the performance in the retro-cue WM task is to analyze the reaction times. I wonder if the authors analyzed it along with the recall error and if it could be informative.

We thank the reviewers for this suggestion. Reaction time in the current context is less informative because accuracy was emphasized during data collection and participants were given a long, fixed time window (3.5 seconds) to render a response based on the method of adjustment. As a result, participants can make multiple button presses when reproducing the remembered orientation gratings, rendering the response time data more complex and harder to interpret (e.g., a big chunk of the RT is the motor response time) than that in the prior research.

2. I would like to make sure if all the presented ROI results were Bonferroni-corrected for multiple comparisons, i.e. if the original α-value was divided by the number of analyses on the dependent variable, here the number of ROIs.

The reviewer is correct that the p values are Bonferroni-corrected through a multiplication of the original p values with the number of tests conducted (see Tables S1 and S2). Please note that minor discrepancy can occur due to rounding. This is equivalent to dividing the α-value by the number of tests conducted (Rosenthal and Rubin, 1983). Here, the number of tests is the number of ROIs.

3. I was wondering if the authors could explain why they selected bilateral ROIs instead of more specific left and right hemisphere regions. Could there be some laterialization effects? Do I understand correctly that the ROI analyses were performed on an average signal from the left and right hemisphere?

Some prior research has suggested that effects related to MTL pattern separation functions can emerge bilaterally (e.g., Montchal et al., 2019; Reagh et al., 2017). As there is not strong prior expectation about lateralized effects, we combine all voxels for bilateral ROIs in our multivariate analyses. We did not perform the analyses separated for each hemisphere and then average the results together, as that would increase the number of tests involved.

4. Could the authors clarify if the representational similarity analysis and IEM were performed on the cued vs. uncued items, regardless of the recall error? The authors performed a nice follow-up control analysis of the IEM reconstructions across precise recall and imprecise recall trails, how about the representational similarity analysis?

Yes, we retained all trials for RSA and IEM analyses, as we clarified in our Method section,

“Under an effective set size of one item, participants’ recall performance was high (Figure 1B), with most recall errors centered around ± 45° of the cued orientation (~97% of the trials) within the ± 90° range. Hence, we retained all trials when investigating the amount of WM information in the recorded neural data during the delay period for multivariate analyses.”

We performed a split-trial analysis based on IEM. This analysis is built upon a prediction-based framework, which allows us to reconstruct the information in each trial of a hold-out test set based on an independent training set from all other trials. That means, the IEM can produce a more balanced trial-level estimate (i.e., reconstruction of remembered information channel) through multiple iterations to allow subsequent averaging or grouping across trials. However, this is not feasible or less efficient based on the trial-wise representational analysis, considering that it is a correlational measure. The magnitude and variability of a correlational measure will be confounded by the number of trials used in the analysis (Schönbrodt and Perugini, 2013). As a result, it is conceptually and analytically challenging to address this confound when the number of smaller- and larger-error trials are too different in RSA.

5. Related to the control analysis of the IEM reconstructions across precise recall and imprecise recall trails – could the authors explain why did they combine data from aLEC and DG/CA3 for this analysis, if in all the other analyses they analyse these ROIs separately?

As we have shown that both the aLEC and DG/CA3 retain a statistically significant amount of information about the cued item (Figures 2 and 3), we have combined them together to reduce the number of tests involved in subsequent analyses. Below, we have shown the results of the split-trial analysis separately for smaller- and larger-error trials. As demonstrated in Author response image 1, the reconstructed information channel relative to the cued item shows less dispersion for the precise recall trials, as compared with the imprecise recall trials. Here, we did not perform separate statistical tests with post-hoc correction for multiple comparisons, considering that the aLEC-DG/CA3 pathway as a whole has been theoretically implicated in predicting behaviors associated memory fidelity (Aimone et al., 2011; Bakker et al., 2008; Ekstrom and Yonelinas, 2020; Korkki et al., 2021; Leal and Yassa, 2018; Marr, 1971; Reagh and Yassa, 2014; Yassa and Stark, 2011) and empirically shown to retained item-specific WM content (Figures 2 and 3).

**Author response image 1. sa2fig1:** IEM reconstruction based on precise and imprecise recall trials separately for aLEC (top row) and DG/CA3 (bottom row).

6. In the caption of Figure 2B the authors mention only correlating the feature similarity of every two cued items with the neural pattern similarity during the WM delay period, but in the analysis (described in C) the effect sizes were compared with the uncued items. Thus, the authors could mention analyzing the uncued items also in B, for the clarity.

We thank the reviewer for this great suggestion. We have now added this information.

**“**Specifically, we correlated the similarity in evoked neural patterns during the WM delay period separately with the feature similarity of every two cued items and with that of every two uncued items.”

Reviewer #3 (Recommendations for the authors):Recommendations for improving the writing and presentation– Check for consistent use of past tense (especially prevalent in Methods)

Thanks for the suggestion. We have now proofread the revised manuscript to minimize grammatical errors.

– Add a more detailed explanation of what model is fitted on which data during the knee-point thresholding, given that part of the results hinge on separation of high- and low-error trials and therefore the exclusion of faulty trials.

In an earlier version of the manuscript, we attempted the knee-point thresholding heuristic to find high- vs. low-error trials. Although this is a common practice in machine-learning literature, feedback from colleagues has suggested us to use a more intuitive heuristic, for example, based on 3 SD of an empirical distribution. Therefore, in our previous submission, we have already replaced the knee-point heuristic to the 3 SD heuristic, which is detailed in the original submission,

“Next, to separate larger-recall errors based on less precise WM representation from those attributable to attention lapses (deBettencourt et al., 2019), we adopted a widely-used thresholding heuristic to find potentially different categories of data points based on the empirical SD of a distribution. Specifically, in our current data, we first calculated the empirical SD (17.33°) of the aggregated raw recall error distribution from all subjects across 2880 trials (ranging from -90° to +90°), which captures the overall variability in participants’ recall performance without a priori model assumption. We then retained the larger-recall error trials within 20° to 3 SD of the recall error distribution (27 ± 3 trials; Figure 4A).”

– Consider including this review in the introduction, Christophel, T. B., Klink, P. C., Spitzer, B., Roelfsema, P. R., and Haynes, J. D. (2017). The distributed nature of working memory. Trends in cognitive sciences, 21(2), 111-124.

Thanks for the suggestion. We have included this important citation in the introduction.

In the introduction,

“This core mental faculty relies upon distributed brain regions (Christophel et al., 2017; Eriksson et al., 2015), ranging from …”

– P.6 mid-page, note that significance and effect size are not the same.

We thank the reviewer for this cautious note. Given the same sample size, significance level is inversely related to effect size (Rosenthal and Rosnow, 2008). To avoid confusion, we have removed the original interpretation about effect size.

“While the rest of the MTL showed similar patterns, we did not obtain significant evidence in other MTL ROIs following the correction of multiple comparisons,…”

– Consider putting paragraph 2 last since 2 is very speculative. Please make sure your wording does not suggest that your data support the role of MTL in reducing interference in WM.

Thanks for this suggestion. We have revised a few places to avoid this confusion.

– In the discussion consider picking up on concepts from the introduction on why you think you found this effect with simple stimuli whereas other studies claim MTL involvement only for more complex stimuli or high-demand tasks.

Thanks for this suggestion. We have revised our discussion on the empirical contribution of this study. Intuitively, the lack of findings using simple stimuli in the past research could be attributed to a lack of sensitivity of the MTL to simple features or lower task loads. With improved spatial resolution in MTL recording, our data may be more sensitive to pick up the representation of simple stimulus features. We have also discussed and tested why some past research using the same paradigm as us has not discovered MTL’s relevance for WM representation.

In Discussion,

“Previously, MTL activity has been shown to scale with WM set size of letters and color squares without decodable item-specific WM … These observations raise the conceptual question concerning the extent to which the MTL responds to task difficulty or retains item-level information in WM. In other words, is the MTL not sensitive to simple stimuli or lower task demands at all? Here, with improved spatial resolution in MTL recordings and using a simple stimulus feature, our data suggest that … These results, therefore, highlight the importance of fine-grained MTL signals in revealing item-specific WM content.”

[Editors' note: further revisions were suggested prior to acceptance, as described below.]

Reviewer #1 (Recommendations for the authors):The authors responded to most of my comments. Some concerns arise as a result of the revisions. Neither in the manuscript nor in the response the authors specified how they ensured assumptions of statistical tests are met. I do appreciate that their results are consistent when using non-parametric alternatives but it would be important to report these results in the manuscript.I suggest to either explicitly test assumptions and wherever those are not met using non-parametric alternatives or skip the parametric statistics and use non-parametric tests to start with. Otherwise, the authors risk reporting tests that are not valid, particularly given their relatively small sample size.

We thank the reviewer for this cautious note. We agree with the reviewer about the importance of ensuring statistical conclusion validity. However, it should also be noted that the common two-stage approach to first check statistical test assumptions and then evaluate the test outcome is not without controversy (García-Pérez, 2012; Wells and Hintze, 2007). Although we had initially reported only non-parametric results, some colleagues have suggested against this approach after our preprint was published online. To balance these different concerns and since our results hold regardless of which statistical approach is adopted, we therefore now report both the parametric and non-parametric test results in the main text. While we understand that there is no “the best” approach to address most statistical issues, it is the convergence of statistical conclusions from different approaches that adds to the solidity of a research finding.

In the Data Analysis section, we added,

“We evaluated statistical significance based on conventional within-subject statistical procedures, such as paired-sample t-tests, with two-tailed p values. Similar results were obtained and verified based on non-parametric statistics (e.g., bootstrapped p values) that have few analytical assumptions (Good, 2013). In particular, we resampled participants’ data with replacement over 1,000 iterations and calculated the empirical two-tailed p values (note: these p values can slightly vary across different iterations of resampling and those smaller than 0.001 are marked as p_bootstrap_ < 0.001).”

In the main text, we reported the p_boostrap_ value associated with each statistical test. Please see the main text for further details.

References

García-Pérez, M. A. (2012). Statistical conclusion validity: Some common threats and simple remedies. Frontiers in Psychology, 3, 1–11. https://doi.org/10.3389/fpsyg.2012.00325

Good, P. (2013). Permutation tests: A practical guide to resampling methods for testing hypotheses. New York, NY: Springer Science & Business Media.

Wells, C. S., & Hintze, J. M. (2007). Dealing with assumptions underlying statistical tests. Psychology in the Schools, 44(5), 495–502. https://doi.org/10.1002/pits.20241